# Studying mixed-species biofilms of *Candida albicans* and *Staphylococcus aureus* using evolutionary game theory

**Sybille Dühring**[ORCID]\*, **Stefan Schuster**

Department of Bioinformatics, Friedrich-Schiller-University Jena, Jena, Germany

\* sybille.duehring@uni-jena.de

## Abstract

Mixed-species biofilms of *Candida albicans* and *Staphylococcus aureus* pose a significant clinical challenge due to their resistance to the human immune system and antimicrobial therapy. Using evolutionary game theory and nonlinear dynamics, we analyse the complex interactions between these organisms to understand their coexistence in the human host. We determine the Nash equilibria and evolutionary stable strategies of the game between *C. albicans* and *S. aureus* and point out different states of the mixed-species biofilm. Using replicator equations we study the fungal-bacterial interactions on a population level. Our focus is on the influence of available nutrients and the quorum sensing molecule farnesol, including the potential therapeutic use of artificially added farnesol. We also investigate the impact of the suggested scavenging of *C. albicans* hyphae by *S. aureus*. Contrary to common assumptions, we confirm the hypothesis that under certain conditions, mixed-species biofilms are not universally beneficial. Instead, different Nash equilibria occur depending on encountered conditions (*i.e.* varying farnesol levels, either produced by *C. albicans* or artificially added), including antagonism. We further show that the suggested scavenging of *C. albicans*' hyphae by *S. aureus* does not influence the overall outcome of the game. Moreover, artificially added farnesol strongly affects the dynamics of the game, although its use as a medical adjuvant (add-on medication) may pose challenges.

## Introduction

Biofilm infections, especially those of fungal and polymicrobial origin, are a significant clinical challenge [1–3]. One of the most serious problems concerning fungal biofilms is their recalcitrance to antifungal therapy [2, 4] as they provide resistance to chemical and physical removal and to antimicrobial drugs [5–7]. Mixed bacterial-fungal biofilms are an even more complex challenge to the immune system and are associated with worse clinical outcomes [8, 9]. In these biofilms, both pathogens become more virulent [10] and the infections are associated with higher morbidity and mortality compared to their mono-infections [11–13]. Using evolutionary game theory (EGT) we study the interactions between the fungus *Candida albicans* and the bacterium *Staphylococcus aureus* in a mixed-species biofilm.

**Data Availability Statement:** All relevant data are within the paper and its Supporting information files.

**Funding:** We acknowledge support by the Deutsche Forschungsgemeinschaft (DFG) within

the CRC/Transregio 124 'Pathogenic fungi and their human host' (project number 210879364). We further acknowledge support by the German Research Foundation Projekt-Nr. 512648189 and the Open Access Publication Fund of the Thueringer Universitaets- und Landesbibliothek Jena. The funders had no role in study design, data collection and analysis, decision to publish, or preparation of the manuscript. There was no additional (internal and external) funding received for this study.

**Competing interests:** The authors have declared that no competing interests exist.

## *C. albicans* and *S. aureus* co-infections and biofilm structure

Together these pathogens can form complex polymicrobial biofilms inside the human host [14–16]. This biofilm formation is usually thought to be mutually beneficial to the fungus and the bacterium as it provides protection [9, 11], metabolic cooperation [16], the ability to colonise and cause infections [17, 18], host evasion [10, 17, 19], and a reduction in drug susceptibility [5–7, 15, 20–24].

As a dimorphic fungus *C. albicans* can change its morphology from yeast to hyphal forms and back [12, 16]. Within a *C. albicans-S. aureus* biofilm one can find both, *C. albicans* yeast and hyphal cells, embedded within an extracellular matrix [25–28]. When interacting with *C. albicans*, *S. aureus* can colonise niches in the host that have otherwise unfavourable conditions and are typically not colonised by the bacterium alone [9, 29]. Even when injected at different body sites of the same host, *S. aureus* disseminates to sites of *C. albicans* infections and settles within the filamentous networks of the *C. albicans* biofilms [9, 11, 13]. In that process, *S. aureus* prefers *C. albicans* hyphae for attachment as well as biofilm formation [20, 23, 30, 31].

## Growth rates and nutrient dependency

In poly-microbial biofilms with *C. albicans*, *S. aureus'* growth is enhanced, resulting in larger biofilms [14, 15, 17]. However, literature findings regarding *C. albicans'* growth in such biofilms are inconclusive. Some studies indicate no significant difference in *C. albicans'* growth between mono- and poly-microbial biofilms [9, 17, 20], while others observe mutual benefit with enhanced growth rates for both species [5, 11, 12].

Whether growing in a mixed-species biofilm together with *S. aureus* is beneficial to *C. albicans* (deduced from an increase in growth rate) may depend on the availability of nutrients in the surrounding environment. In settings where *S. aureus* and *C. albicans* compete for nutrients [11] and as the biofilm matures, the number of dead *C. albicans* cells increases [15]. It is commonly assumed that poly-microbial biofilms do not exhibit lethality towards either organism, allowing *C. albicans* and *S. aureus* to grow without antagonism [16, 17]. However, Shirtliff *et al.* [22] suggest that the initially synergistic relationship between *C. albicans* and *S. aureus* can transition to competition or even antagonism at a certain point [22] and it was even suggested that under sparse nutritional conditions *S. aureus* can scavenge *C. albicans'* cell wall [15].

## Farnesol as a quorum sensing molecule of *C. albicans*

As a protective strategy against bacterial attacks and nutrient competition, *C. albicans* employs farnesol [32, 33]. Farnesol, an acyclic sequestering alcohol, acts as a quorum sensing molecule in *C. albicans*, regulating biofilm formation by inhibiting the yeast-to-hyphae transition [28, 32, 34]. This transition is crucial for tissue invasion and biofilm development [11]. High farnesol levels inhibit filamentation and promote yeast budding in *C. albicans* [35, 36]. By regulating hyphae formation in dense cell populations like biofilms, *C. albicans* gains protection against bacterial competitors and the host immune system [32]. Farnesol also prevents Nrg1 degradation in *C. albicans* [37], facilitating the dispersion of yeast cells from mature biofilms and colonisation of new sites [28]. Additionally, farnesol quorum sensing is believed to conserve energy for *C. albicans* when nutrients get sparse [32, 33].

Farnesol not only protects *C. albicans* by influencing its morphology, but it also affects *S. aureus* in various ways. It impairs *S. aureus'* biofilm formation and compromises its cell membrane integrity, viability, and susceptibility to antibiotics [6, 9, 22]. Farnesol inhibits *S. aureus'* growth and reduces its biofilm formation. It increases the bacterium's sensitivity to antibiotics, particularly in highly resistant *S. aureus* strains [11, 23]. That effect may be attributed to cell

membrane damage and improved antibiotic diffusion to target sites [9]. The impact of *C. albicans'* farnesol on *S. aureus'* biofilm development depends on the concentration of farnesol [2], with the highest vancomycin tolerance observed at levels of 30 μM to 40 μM [38]. Kong *et al.* [38] propose that farnesol induces oxidative stress, activating a protective stress response through efflux pumps in *S. aureus*, which subsequently leads to heightened tolerance against antibacterials [38]. Thus, *S. aureus'* tolerance to antimicrobials initially increases with low concentrations of farnesol before dramatically decreasing at higher concentrations [38]. High farnesol concentrations result in a decrease in *C. albicans'* hyphae, as well as the viability and biofilm capability of *S. aureus*, ultimately leading to the end of the mixed-species biofilm. This suggests a potential role for *C. albicans'* farnesol in orchestrating interactions within the mixed biofilm of *C. albicans* and *S. aureus* [7, 22].

## Analysing biofilm dynamics using evolutionary game theory

Even though cooperation between species is evolutionary harder to explain than the cooperation of the same species or genotype, polymicrobial biofilms provide the basic conditions that enable cooperation between different species. Certain conditions can lead to cooperation in biofilms, benefiting all species involved. Characterising interactions within a complex system as either cooperation or competition requires careful consideration of multiple factors. Even though the actual biomass might remain unchanged or decrease, one species can benefit by gaining protection from environmental stressors or expanding into new niches, ultimately resulting in an increased overall fitness [10, 39].

To study the complex interactions between species EGT provides a helpful tool and conceptual framework. EGT, especially Nash equilibria and replicator equations are applied to many research fields, such as ecology, medicine, bacterial populations and cancer therapy [40–48]. For an introduction to EGT and replicator equations as well as their applications to microbial biology see [49–51]. For studies on the host-pathogen interactions of *C. albicans* and the human immune system, using EGT see [52–54].

Hummert *et al.* [52] suggest using game theory to study the morphological transition of *C. albicans* inside human macrophages. They establish a symmetric game between *C. albicans* cells, including the human host as a third player by representing it through adjustable parameters. Tyc *et al.* [55] use a symmetrical two-player game to model the pair-wise interactions of *C. albicans* yeast and hyphae cells, analysing the morphological changes in *C. albicans*. They assume the player's payoffs to be nutrient-dependent and use sigmoidal population dynamics as payoff functions. They further derive replicator equations for the fungal population from the payoff matrix of their symmetric game to infer the dynamics of the yeast to hyphae ratio within a *C. albicans* population.

In this paper, we use EGT to analyse the complex interactions of *C. albicans* and *S. aureus* in a mixed-species biofilm. For this we extend the symmetrical approaches proposed by Hummert *et al.* [52] and Tyc *et al.* [55] to an asymmetrical two-player game between *S. aureus* and *C. albicans*. In the Methods section, we establish the payoff matrices for both players, explaining the benefits and costs for each strategy pairing, and derive the resulting replicator equations of the game. In the Results section we study the biofilm interactions on a population level and analyse the stability conditions for different population profiles. By determining the fixed points of the replicator equations we establish the Nash equilibria of the game and extend our studies to evolutionary stable strategies. We study the qualitative behaviour of the biofilm dynamics by simulating possible population profiles under different parameter conditions. We incorporate variable nutrient and farnesol levels to model the variations in nutrient abundance and scarcity in the environment, as well as the impact of farnesol on the system. As it is

suggested that *S. aureus* can switch from cooperation to exploitation and feed on *C. albicans* hyphae [15] we distinguish two cases. In the first case, we consider the feeding of *C. albicans'* hyphae by *S. aureus* with a resulting benefit for the bacterium and loss for the fungus, whereas in the second case, no such feeding occurs.

## Methods

### Characterisation of the game

Using EGT, we study the (pairwise) interactions between the fungus *C. albicans* and the bacterium *S. aureus* in a mixed-species biofilm. Each player can adopt two strategies. For *S. aureus* we distinguish between the strategies "cooperation" (coo) and "exploitation" (exp). For *C. albicans* we distinguish between the strategies "yeast" (ye) and "hyphae" (hy). Each cell's payoff, denoted by

$$E_{\text{player}}(C.\ albicans'\ \text{strategy},\ S.\ aureus'\ \text{strategy}),$$

is determined by its own and the other player's choice in strategy.

When a *S. aureus* cell adopting the cooperative strategy encounters a *C. albicans* cell in its yeast form we assume that they neither harm nor benefit each other, as *S. aureus* prefers *C. albicans* hyphae to attach to [14, 15, 20, 23, 30, 31]. Hence both players gain their basic fitness as a benefit. To model each player's basic fitness we follow Tyc *et al.*'s [55] assumption of sigmoidal population growth for *C. albicans* yeast ($E_Y$) and hyphae ($E_H$) and extend it to *S. aureus* ($E_S$). The payoffs of *C. albicans* yeast and *S. aureus* cooperators are denoted as:

$$
\begin{aligned}
E_{C.albicans}(\text{ye},\ \text{coo}) &= E_Y = \frac{u_Y n^\sigma}{u_Y + n^\sigma} \quad \text{and} \\[2mm]
E_{S.aureus}(\text{ye},\ \text{coo}) &= E_S = \frac{u_S n^\sigma}{u_S + n^\sigma}.
\end{aligned}
\tag{1}
$$

The nutrition provided by the environment is denoted by *n* while *u* and *σ* are morphology dependent parameters.

To gain more nutrients from the host, *C. albicans* can change its morphology from yeast to hyphae [56]. This change in morphology induces the mixed-species biofilm formation of *C. albicans* and *S. aureus* [16]. When a *S. aureus* cell adopting the cooperative strategy is interacting with a *C. albicans* hyphae cell the invasion of host tissue, driven by *C. albicans*, is simplified through *S. aureus* virulence factors [18]. Both players' gain by invading the host for nutrients is denoted by $I_1$ and is shared between *S. aureus* and *C. albicans* hyphae with factor *b*. Additionally, the resistance of the mixed-species biofilm is significantly increased compared to the resistance of both single-species biofilms [6, 23, 24]. This increase in resistance is denoted by the factor *r* and affects both *S. aureus* and *C. albicans* hyphae. The payoffs of the two players are then:

$$
\begin{aligned}
E_{S.aureus}(\text{hy},\ \text{coo}) &= rE_S + bI_1 \quad \text{and} \\[2mm]
E_{C.albicans}(\text{hy},\ \text{coo}) &= rE_H + (1-b)I_1 \quad \text{with} \\[2mm]
E_H &= \frac{u_H n^\sigma}{u_H + n^\sigma}
\end{aligned}
\tag{2}
$$

as *C. albicans'* hyphae basic fitness.

Once the nutrients provided by the host are running low it is suggested that *S. aureus* can switch from cooperation to exploitation and feed on *C. albicans* hyphae [15]. To test this

hypothesis we denote the gain and loss resulting from the theft of nutrients originating from *C. albicans* hyphae by $I_2$. For simplicity, we set $I_1 = 0$ and assume that the host no longer provides any nutrients. Allowing for very small values of $I_1$ might be an interesting future study. The potential parasitism of *S. aureus* can be countered by *C. albicans* with farnesol [32, 33]. The costs for the farnesol production by *C. albicans* are denoted by $f_1$.

Farnesol affects *S. aureus* in more than one way. Firstly, the *C. albicans* hyphae population is gradually decreasing and, thus, so are *S. aureus'* potential exploitation partners. As yeast cells remain in their yeast form and hyphae cells die through *S. aureus'* exploitation, the bacteria cells encounter fewer hyphae cells to feed on. Secondly, farnesol is impacting the growth of *S. aureus* [6, 23, 38]. This change in growth is considered by $f_2$. While very low levels of farnesol can have a positive effect on *S. aureus'* growth by causing a heightened tolerance against antibacterials ($f_2 < 0$ increasing the growth rate), increasing amounts of farnesol negatively affect *S. aureus'* growth as the tolerance against antibacterials drastically drops ($f_2 > 0$ decreasing the growth rate) [38]. The payoffs for the two players in this stage of the biofilm are:

$$E_{C.albicans}(\text{hy, exp}) = rE_H - I_2 - f_1 \quad \text{and}$$
$$E_{S.aureus}(\text{hy, exp}) = rE_S + I_2 - f_2.$$

(3)

As the release of farnesol promotes *C. albicans'* yeast form, a *S. aureus* cell adopting the exploitation strategy is encountering more and more *C. albicans* yeast cells. This usually marks the end of the mixed-species biofilm. The payoffs for both players in this scenario are:

$$E_{C.albicans}(\text{ye, exp}) = E_Y - f_1 \quad \text{and}$$
$$E_{S.aureus}(\text{ye, exp}) = E_S - f_2.$$

(4)

Farnesol is suggested as a medical adjuvant (add-on medication) to shift the balance of a mixed-species biofilm to more favourable conditions for the patients [57]. To test the impact of artificially added farnesol on the system we introduce $f_{ar}$ as the change in growth caused by artificially added farnesol. We assume that the concentrations of the artificially added farnesol are always high enough, that $f_{ar}$ is reducing *S. aureus* growth. Therefore, we subtract $f_{ar}$ from all *S. aureus'* payoffs. High concentrations of farnesol also reduce *C. albicans* hyphae's growth. Hence, we subtract $f_{ar}$ from $E_{C.\,albicans}(\text{hy, coo})$ and $E_{C.\,albicans}(\text{hy, exp})$. For *C. albicans* yeast cells the effect of added farnesol depends on the opponent they are playing against. For *C. albicans* yeast cells playing against cooperating *S. aureus* cells, we assume no effect of the added farnesol on $E_{C.\,albicans}(\text{ye, coo})$. For *C. albicans* yeast cells playing against exploiting *S. aureus* cells, we assume a positive effect as *C. albicans'* farnesol production costs are lowered. Hence, we add $f_{ar}$ to $E_{C.\,albicans}(\text{ye, exp})$. All variables used to model the game are given in Table 1. A summary of the four considered biofilm stages of the game and the payoffs each player gains at each stage is given in Fig 1.

## The replicator equation model

To study the considered fungal-bacteria interactions on a population level we use replicator equations. Replicator equations are deterministic, monotonic, nonlinear differential equations. They are used in EGT to describe the dynamics of populations in which successful individuals outnumber less successful individuals. Each strategy of the game (described in the Methods section) is "played" by a certain fraction of the population. Let $x \in [0, 1]$ be the fraction of *C. albicans* yeast cells in the fungal population. Then $1 - x$ describes the fraction of *C. albicans* hyphal cells in the fungal population. For the *S. aureus* population, $y \in [0, 1]$ denotes the fraction of cooperating cells, while $1 - y$ gives the fraction of the *S. aureus* cells which have adopted

**Table 1. List of all variables, their description and domain of definition used to model the interactions of *C. albicans* and *S. aureus* in a mixed-species biofilm as an evolutionary game.**

| Variable | Description | Domain of definition |
|---|---|---|
| $E_Y$ | Basic fitness of *C. albicans* yeast cells | $E_Y \in \mathbb{R}^+$ |
| $E_H$ | Basic fitness of *C. albicans* hyphae cells | $E_H \in \mathbb{R}^+$ |
| $E_S$ | Basic fitness of *S. aureus* cells | $E_S \in \mathbb{R}^+$ |
| $n$ | Nutrients provided by the environment | $n \in \mathbb{R}^+$ |
| $u$ and $\sigma$ | Morphology dependent parameters defining the nutrient uptake | $u_Y, u_S, u_H, \sigma \in \mathbb{R}^+$ |
| $r$ | Increase in resistance | $r \in \mathbb{R}^+$ |
| $b$ | Factor indicating the share in nutrients | $b \in (0, 1)$ |
| $I_1$ | Both players' gain by invading the host for nutrients | $I_1 \in \mathbb{R}^+$ and hence $(1 - b)I_1 \in \mathbb{R}^+$ |
| $I_2$ | Gain and loss (depending on the player) resulting from the theft of nutrients originating from *C. albicans* hyphae | $I_2 \in \mathbb{R}_0^+$ |
| $f_1$ | Farnesol production costs | $f_1 \in \mathbb{R}^+$ |
| $f_2$ | Change in growth caused by released farnesol | $f_2 \in \mathbb{R}$ |
| $f_{ar}$ | Change in growth caused by artificially added farnesol | $f_{ar} \in \mathbb{R}_0^+$ |
| $x$ | Fraction of *C. albicans* yeast cells in the fungal population | $x \in [0, 1]$ |
| $y$ | Fraction of *S. aureus* cooperating cells in the bacterial population | $y \in [0, 1]$ |

the exploitation strategy. A population profile of the *C. albicans* population can then be given by the strategy vector $\boldsymbol{x} = (x, 1 - x)^T$. Equivalently the population profile of the *S. aureus* population is given by the strategy vector $\boldsymbol{y} = (y, 1 - y)^T$. The payoff matrices (established in the Methods section) for both players are

$$A = \begin{pmatrix} E_Y & E_Y - f_1 + f_{ar} \\ rE_H + (1 - b)I_1 - f_{ar} & rE_H - I_2 - f_1 - f_{ar} \end{pmatrix} \qquad (5)$$

for the *C. albicans* population and

$$B = \begin{pmatrix} E_S - f_{ar} & rE_S + bI_1 - f_{ar} \\ E_S - f_2 - f_{ar} & rE_S + I_2 - f_2 - f_{ar} \end{pmatrix} \qquad (6)$$

for the *S. aureus* population. The replicator equations for both players are given by

$$\dot{x} = x((A\boldsymbol{y})_1 - \boldsymbol{x}^T A \boldsymbol{y}) \qquad \text{and}$$

$$\dot{y} = y((B\boldsymbol{x})_1 - \boldsymbol{y}^T B \boldsymbol{x}) \qquad (7)$$

with $(A\boldsymbol{y})_1$ and $(B\boldsymbol{x})_1$ being the expected payoffs for $x$ and $y$ and $\boldsymbol{x}^T A \boldsymbol{y}$ and $\boldsymbol{y}^T B \boldsymbol{x}$ being the mean payoffs of the *C. albicans* and *S. aureus* populations. From this, we derive the replicator equations

$$\dot{x} = x(1 - x)(E_Y - rE_H + I_2 + 2f_{ar} - y((1 - b)I_1 + I_2 + f_{ar})) \qquad \text{and}$$

$$\dot{y} = y(1 - y)(bI_1 - I_2 + f_2 + x(I_2 - bI_1)), \qquad (8)$$

describing the dynamic changes of both population profiles over time. To test the hypothesis of *S. aureus* feeding on *C. albicans* we either assign a positive value to $I_2$, in case of exploitation, or set $I_2$ equal to 0, in case of no exploitation. To evaluate the two cases of no and added

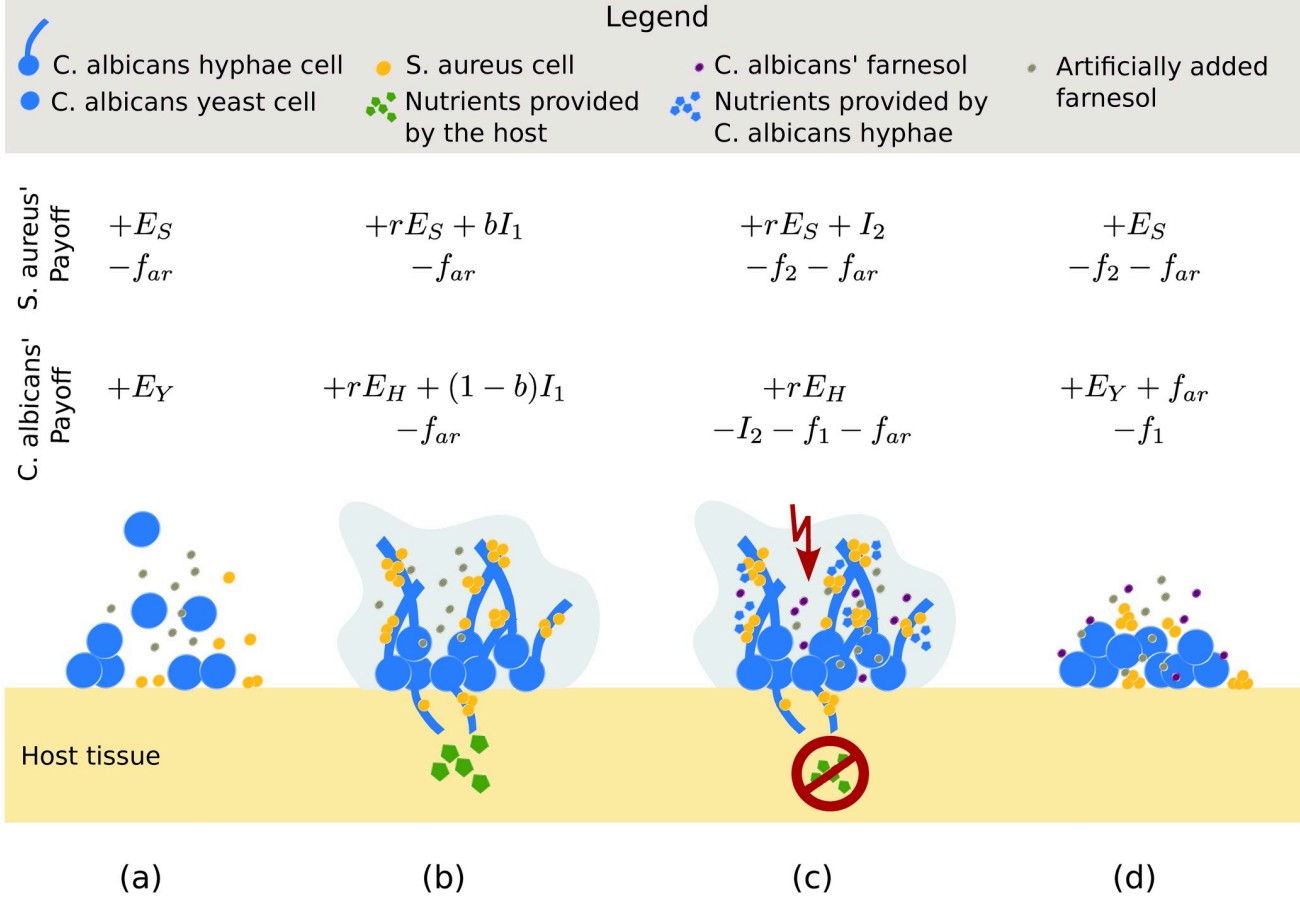

**Fig 1. The four stages of the biofilm with the resulting payoffs for each pathogen within the evolutionary game.** a) Initial stage: *S. aureus* cells using the cooperative strategy are encountering *C. albicans* yeast cells (with yeast strategy). Both players gain their basic fitness ($E_S$ and $E_Y$) as a benefit. If artificially added farnesol is present in the biofilm surroundings the change in growth it is causing is given by $f_{ar}$ in the payoffs. b) Beginning of the biofilm: *S. aureus* cells with the cooperative strategy and *C. albicans* cells with hyphae strategy are forming a biofilm. Both pathogens' gain in fitness ($E_S$ and $E_H$) is increased by $r$ due to the increase in biofilm resistance. Both players' gain by invading the host for nutrients is denoted by $I_1$ and is shared between *S. aureus* and *C. albicans* hyphae with the factor $b$. c) Disturbance of the biofilm: With low or no amounts of nutrients from the host *S. aureus* (with exploitation strategy) is feeding on *C. albicans* hyphae (with hyphae strategy). The gain and loss resulting from the theft of nutrients originating from *C. albicans* is given by $I_2$. As a defence *C. albicans* starts releasing its own farnesol. The costs for the farnesol production by *C. albicans* are denoted by $f_1$. The effects of the released farnesol on *S. aureus* are denoted by $f_2$. d) End of the biofilm: Due to the released farnesol *S. aureus* cells with exploitation strategy are encountering *C. albicans* cells with yeast strategy.

farnesol we either assign a positive value, in case of artificially added farnesol, or 0, in case of no farnesol addition, to $f_{ar}$.

## Results

### Nash equilibria and fixed points

One solution concept of a game is the Nash equilibrium concept. A Nash equilibrium is a set of strategies, one for each player of the game, where none of the players has an incentive to switch strategy unilaterally [58, 59]. It can be interpreted as a player's best response to the strategy choice of the other players [60–62]. It can be shown that the Nash equilibria of a game are the asymptotically stable fixed points of the replicator equation model of the same game [49–51]. To analyse the dynamics of the game we are looking for solutions to $\dot{x} = \dot{y} = 0$. We find

five fixed points $(x^*, y^*)$, including the four trivial solutions $x^*, y^* \in \{0, 1\}$ (that is $(0,0)$, $(0,1)$, $(1,0)$ and $(1,1)$) and for $x^*, y^* \in (0, 1)$, a mixed strategy solution

$$(\hat{x}, \hat{y}) = \left( \frac{bI_1 - I_2 + f_2}{bI_1 - I_2}, \frac{E_Y - rE_H + I_2 + 2f_{ar}}{I_2 + (1 - b)I_1 + f_{ar}} \right), \tag{9}$$

provided that

$$0 < \frac{bI_1 - I_2 + f_2}{bI_1 - I_2} < 1 \quad \text{and}$$

$$0 < E_Y - rE_H + I_2 + 2f_{ar} < I_2 + (1 - b)I_1 + f_{ar}. \tag{10}$$

By performing simulations for different parameter sets, we can get an idea of how the populations interact over time under different conditions. In this way, mathematical modelling can support experimental studies. Ideally, the interplay between computational biology studies and wet lab experiments fosters an improved understanding of these complex systems [63, 64]. For the system under study, quantitative experimental data are scarce so far. However, our aim is to provide a framework for hypothesis generation and verification within a qualitative study. Our simulations serve to show the general behaviour of the system. That is why we have flexibility in selecting representative values for the simulation as long as they adhere to the stability conditions corresponding to the four pure Nash equilibria of the game. It is worth mentioning that for determining the pure Nash equilibria, only the order relations among the payoffs rather than their exact values are relevant.

Fig 2 shows possible population profiles of the game over time under varying parameter conditions given in Table 2. For the simulations, we assumed no interference from third parties and set the artificial farnesol level $f_{ar}$ equal to zero. As morphology-dependent parameters, we fix $u_Y$, $u_H$, $u_S$, and $\sigma$ arbitrarily and assume a variation of $E_Y$, $E_H$, and $E_S$ to be due to a variation in the available nutrients of the environment $n$. For the specific values of $u_Y$, $u_H$, and $\sigma$ we follow Tyc *et al.* [55] and set $u_S$ accordingly.

Depending on the parameter values, we see different outcomes for the strategy fractions of *C. albicans* and *S. aureus*. Our simulations show all five fixed point cases. Each case represents a possible state of the biofilm. Fig 2 shows the simulation results of the case $I_2 \neq 0$. It can be seen in Fig 2c and 2d that the course of the curve and the time needed to reach a fixed point are parameter-dependent. We find that for the inner fixed point oscillations can occur (Fig 2f). The simulations of the case $I_2 = 0$ are very similar reaching the same fixed points. This is the case as the dynamical behaviour of the curves of both cases is qualitatively the same, but reached with different parameter values (for details see the S1 File). As in the case of $I_2 \neq 0$ again in the case of $I_2 = 0$, the course of the curve and the time needed to reach a fixed point can differ as they are parameter-dependent.

For a system to converge to a Nash equilibrium the corresponding fixed point needs to be asymptotically stable [49–51]. Therefore, to derive the Nash equilibria of the game we have to analyse the stability of the fixed points.

## Stability analysis and evolutionary stable strategies

To analyse the stability of the five fixed points we use standard methods from local stability analysis based on the eigenvalues of the Jacobian matrix. For the five fixed points of the system, we find that the fixed points $(0, 0)$, $(1, 0)$, $(0, 1)$ and $(1, 1)$ are under the stability conditions given in Table 3 asymptotically stable and otherwise unstable (for details see the S2 File). In the

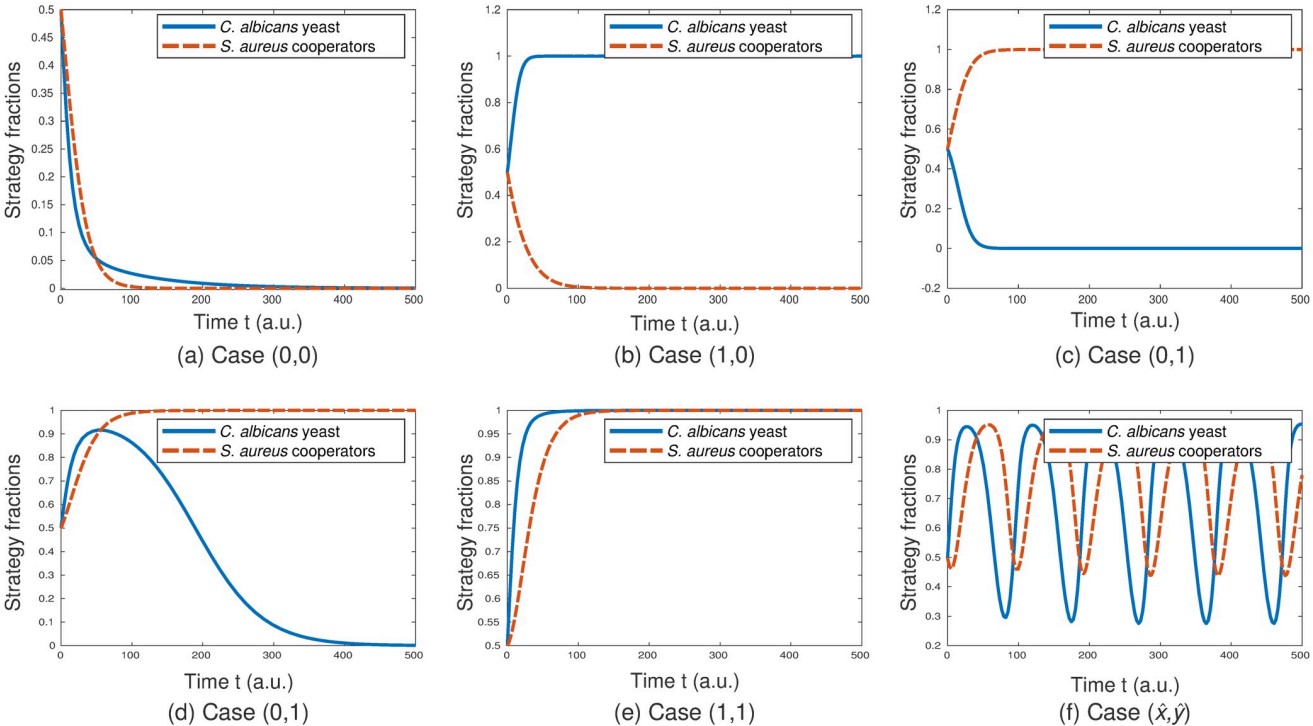

**Fig 2. The dynamical behaviour of the game.** The simulations show possible population profiles of *C. albicans* and *S. aureus* over time for varying parameter settings. For the parameter details of the simulations see Table 2. The end points of the simulations show all five fixed point cases with each case representing a possible state of the biofilm. All six simulations start with a 50/50 mix of hyphae and yeast cells in the *C. albicans* population and a 50/50 mix of exploiting and cooperating cells in the *S. aureus* population. Observations at the end of the simulations: a) All *C. albicans* cells adopt the hyphae strategy and all *S. aureus* cells adopt the exploitation strategy. b) All *C. albicans* cells adopt the yeast strategy and all *S. aureus* cells adopt the exploitation strategy. c) All *C. albicans* cells adopt the hyphae strategy and all *S. aureus* cells adopt the cooperation strategy. d) After an increase in *C. albicans* yeast cells at the beginning of the simulation, all *C. albicans* cells adopt the hyphae strategy and all *S. aureus* cells adopt the cooperation strategy. The simulations 2c and d both end in the same biofilm state. However, the course of the curve and the time needed to reach a fixed point are parameter-dependent. e) All *C. albicans* cells adopt the yeast strategy and all *S. aureus* cells adopt the cooperation strategy. f) For the inner fixed point we see oscillations of the population profiles with all four biofilm stages being possible.

case of stability, the four fixed points are the strict Nash equilibria (in pure strategies) of the game.

It is noteworthy to mention that there is an alternative way to determine the four strict Nash equilibria of the game. This can be done by determining the maximum of each column

**Table 2. Parameter values corresponding to the different cases shown in Fig 2.**

| Parameter | Case (0, 0) (Fig 2a) | Case (1, 0) (Fig 2b) | Case (0, 1) (Fig 2c/Fig 2d) | Case (1, 1) (Fig 2e) | Case $(\hat{x}, \hat{y})$ (Fig 2f) |
|---|---|---|---|---|---|
| $n$ | 0.83 | 1.5 | 0.1 / 1.5 | 1.5 | 0.1 |
| $I_1$ | 0.2 | 0.2 | 0.2 | 0.1 | 0.2 |
| $I_2$ | 0.09 | 0.12 | 0.05 / 0.12 | 0.12 | 0.4 |
| $f_2$ | −0.05 | −0.05 | 0.05 | 0.05 | 0.1 |

For all cases: $u_Y = 2$, $u_H = 1$, $u_S = 2.5$, $\sigma = 3$, $f_1 = 0.02$, $r = 1.5$, $b = 0.4$, $f_{ar} = 0$ with all values listed having arbitrary units.

**Table 3. The four ESS of the game (each representing a stage of the biofilm), their corresponding fixed points and the stability conditions necessary for reaching the fixed points.**

| Fixed Point $(x^*, y^*)$ | Stability Condition | Game Scenario / ESS |
|:---:|:---|:---|
| $(0, 0)$ | $rE_H - E_Y > I_2 + 2f_{ar}$ and $f_2 < I_2 - bI_1$ | Hyphae and Exploitation |
| $(1, 0)$ | $rE_H - E_Y < I_2 + 2f_{ar}$ and $f_2 < 0$ | Yeast and Exploitation |
| $(0, 1)$ | $E_Y - rE_H < (1 - b)I_1 - f_{ar}$ and $f_2 > I_2 - bI_1$ | Hyphae and Cooperation |
| $(1, 1)$ | $E_Y - rE_H > (1 - b)I_1 - f_{ar}$ and $f_2 > 0$ | Yeast and Cooperation |

of the two matrices $A$ (Eq (5)) and $B$ (Eq (6)). For example, if

$$E_Y > rE_H + (1 - b)I_1 - f_{ar} \quad \text{and}$$

$$E_S - f_{ar} > E_S - f_2 - f_{ar} \tag{11}$$

the strategy pair Yeast—Cooperation (corresponding to the fixed point $(1, 1)$) is a Nash equilibrium. Or if

$$rE_H - I_2 - f_1 - f_{ar} > E_Y - f_1 + f_{ar} \quad \text{and}$$

$$rE_S + I_2 - f_2 - f_{ar} > rE_S + bI_1 - f_{ar} \tag{12}$$

the strategy pair Hyphae—Exploitation (corresponding to the fixed point $(0, 0)$) is a Nash equilibrium. Of course, the Nash equilibria and the conditions under which they occur, are the same regardless of the method used.

The eigenvalues of the inner fixed point $(\hat{x}, \hat{y})$ are either real and positive, or purely imaginary (complex conjugated eigenvalues with real part equals 0). In the case of positive eigenvalues, the inner fixed point is unstable. Under specific conditions (for details see the S2 File) pure imaginary eigenvalues can occur, indicating a centre or spiral. In this case, the inner fixed point $(\hat{x}, \hat{y})$ can either be stable or unstable with further analysis needed to determine the stability. However, as we are considering an asymmetric game, only the four strict Nash equilibria can be evolutionary stable strategies (ESS).

First described by Maynard Smith and Price [65], an ESS is a strategy (or strategy vector) that when adopted by all players of a population, no mutant strategy (or alternative best reply to an ESS) can invade. It can be interpreted as the optimal strategy choice for each individual to gain the maximal payoff. Each deviation from this optimal strategy vector leads to a fitness payoff below the mean payoff of the entire population. This self-regulating mechanism stabilises the ratio of strategy frequencies within a population and the strategy choice of individuals. The solution concept of ESS thereby refines the solution concept of the Nash equilibrium by adding a stability requirement regarding deviation or mutant strategies [65, 66]. For asymmetric games, Selten proved that the ESS has to be a strict Nash equilibrium [66]. We, therefore, omit further analysis of the inner fixed point and focus our discussions on the four ESS of the system.

## What conditions influence *C. albicans'* strategy choice?

From the stability conditions in Table 3, we can infer that *C. albicans* adopts the hyphae strategy when the nutrient gain from invading the host $I_1$ and the resistance gain $r$ are high or when the depletion/exploitation by *S. aureus* $I_2$ and the growth change caused by artificially added farnesol $f_{ar}$ are low. Accordingly, we find that high levels of $f_{ar}$ and $I_2$ as well as low levels of $I_1$ and $r$ support *C. albicans'* strategy choice of yeast growth.

## What conditions influence *S. aureus'* strategy choice?

In situations where the yeast strategy of *C. albicans* is favoured, *S. aureus'* strategy choice depends on the growth change caused by released farnesol ($f_2$). If the effect of farnesol is beneficial for *S. aureus* ($f_2 < 0$) the bacterium chooses its exploitation strategy. If, on the contrary, the effect of farnesol is harmful to the bacterium ($f_2 > 0$) *S. aureus* adopts its cooperation strategy (see Table 3). While there are still *S. aureus* exploiters in *C. albicans'* surroundings the fungus does well to increase the amount of farnesol. Increases in farnesol concentrations are likely to cause a strategy shift from exploitation to cooperation in *S. aureus*, as low concentrations of farnesol have a beneficial effect on *S. aureus*, while higher doses have a detrimental effect [38].

If *C. albicans* chooses its hyphae strategy *S. aureus'* strategy choice depends on the relation between the change in growth caused by released farnesol and the gain in nutrients by feeding on *C. albicans* instead of the host ($f_2 \gtreqless I_2 - bI_1$). Low or even beneficial effects of farnesol on *S. aureus* ($f_2$), low levels of nutrients provided by the host ($I_1$), and a sufficiently high gain in nutrients through feeding on *C. albicans* ($I_2$) cause *S. aureus* to choose its exploitation strategy. If instead, the level of nutrients provided by the host ($I_1$) or the effect of farnesol on *S. aureus* ($f_2$) are high *S. aureus* chooses cooperation.

It is noteworthy to recognise that high levels of nutrients provided by the host should make it unnecessary for *S. aureus* to feed on *C. albicans* risking the benefits of cooperation. In the case of equal levels of nutrients supplied by the host and *C. albicans*, the bacterium's choice in strategy again only depends on the effect of farnesol on *S. aureus*. Fig 3 provides a summary of the conditions influencing the strategy choices of *C. albicans* and *S. aureus*, as well as the resulting ESS representing different stages of the biofilm.

## The influence of artificially added farnesol ($f_{ar}$) and nutritional exploitation by *S. aureus* ($I_2$) on the system

To study the influence of nutrients ($n$) and the change in growth caused by farnesol ($f_{ar}$ and $f_2$) on the mixed-species biofilm, we calculate the strategy fractions $x$ and $y$ at time step 100 (a.u.), where a quasi-equilibrium has already been established. We set $I_2 = 0.12$ to simulate the case of exploitation or $I_2 = 0$ in case of no exploitation. The parameter values of $u_Y$, $u_H$, $u_S$, $\sigma$, $f_1$, $r$, and $b$ are given in Table 2. We assume that $I_1 = 0.2$ and simulate over $n \in [0, 5]$, $f_{ar} \in [0, 0.1]$ and $f_2 \in [-0.1, 0.1]$. The results are shown for selected $f_2$ levels in Fig 4 (with $I_2 \neq 0$) and Fig 5 (with $I_2 = 0$).

We find different population profiles of *C. albicans* and *S. aureus* depending on the number of nutrients ($n$), the levels of change in growth caused by farnesol ($f_{ar}$ and $f_2$) and the gain and loss resulting from the exploitation ($I_2$). To summarise the results for both populations in Table 4 we distinguish between the two cases of exploitation ($I_2 \neq 0$) and no exploitation ($I_2 = 0$). For each case, we distinguish between different degrees of severity (beneficial, unsusceptible, low, moderate and high levels of change) in which released farnesol ($f_2$) is affecting the *S. aureus* population. We further distinguish between different levels of change (high and low) caused by artificially added farnesol ($f_{ar}$).

### The case of $I_2 \neq 0$

Low farnesol concentrations can enhance the growth of *S. aureus* [38]. If farnesol levels are beneficial for the bacterium ($f_2 < 0$), we only observe yeast in the *C. albicans* population and exploiters in the *S. aureus* population independent of $n$ or $f_{ar}$ (for details see the S1 File).

For unsusceptible *S. aureus* strains ($f_2 = 0$) only yeasts are observed in the *C. albicans* population independent of $n$ or $f_{ar}$. The population profile of *S. aureus* depends on the levels of $f_{ar}$

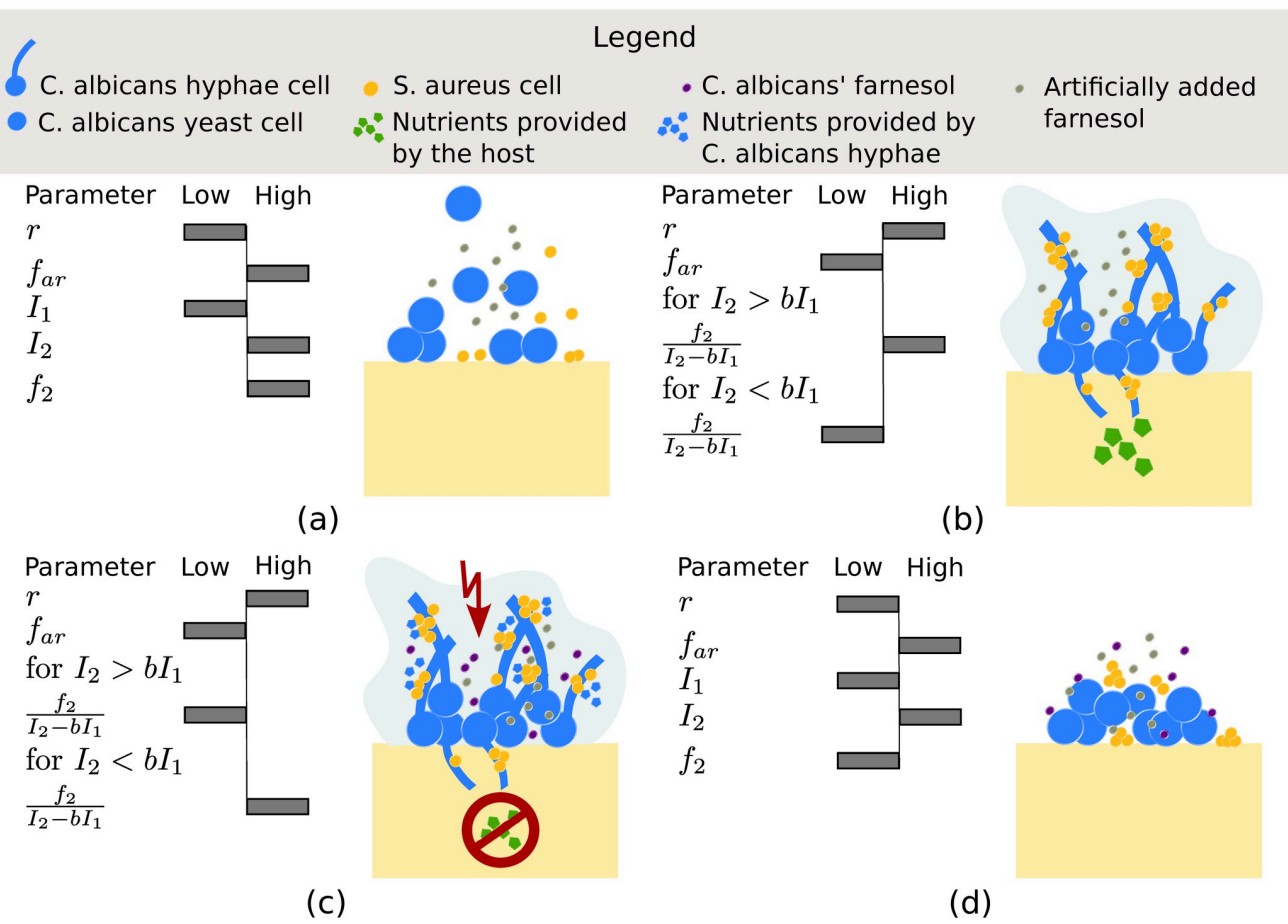

**Fig 3. The four ESS of our game (each representing a stage of the biofilm) and the parameter conditions influencing the strategy choice of each pathogen.** a) Yeast and Cooperation: For low $I_1$ and $r$ values and high $I_2$, $f_{ar}$ and $f_2$ values, *C. albicans* chooses its yeast strategy while *S. aureus* chooses its cooperation strategy. b) Hyphae and Cooperation and c) Hyphae and Exploitation: For high $r$ and low $f_{ar}$ values the outcome of the game depends on the relation of $f_2 \gtreqless I_2 - bI_1$. For $I_2 > bI_1$ and $\frac{f_2}{I_2 - bI_1} > 1$ as well as for $I_2 < bI_1$ and $\frac{f_2}{I_2 - bI_1} < 1$ *C. albicans* chooses its hyphae strategy while *S. aureus* chooses its cooperation strategy (Fig 3b). When instead $I_2 > bI_1$ and $\frac{f_2}{I_2 - bI_1} < 1$ as well as for $I_2 < bI_1$ and $\frac{f_2}{I_2 - bI_1} > 1$ *C. albicans* chooses its hyphae strategy while *S. aureus* chooses its exploitation strategy (Fig 3c). d) Yeast and Exploitation: For low $I_1$, $r$ and $f_2$ values but high $I_2$ and $f_{ar}$ values, *C. albicans* chooses its yeast strategy while *S. aureus* chooses its exploitation strategy.

(see Fig 4c). High levels of $f_{ar}$ lead to a mixed population profile of cooperators and exploiters. For low levels of $f_{ar}$, we find a groove towards exploiters reaching zero levels for low amounts of nutrients ($n$). The depth of the groove is nutrient-dependent and levels out for higher $n$ values.

For low levels of $f_2$ ($f_2 = 0.02$), both population profiles are strongly affected by the artificial farnesol $f_{ar}$ (see Fig 4a and 4d). For high levels of $f_{ar}$ ($f_{ar} > 0.035$), the population profile of *C. albicans* consists of yeast while the population profile of *S. aureus* consists of cooperators. However, for low levels of $f_{ar}$ ($f_{ar} < 0.035$) the strategy fractions of both populations show irregular oscillations. We consider these oscillations to be the outcome of the unstable inner fixed point's oscillations, rather than numerical artefacts.

Moderate to high levels of $f_2$ ($f_2 \geq 0.04$), cause a switch in the *C. albicans* population profile at $f_{ar} = 0.035$ similar to a bang-bang control. This results in an immediate change from hyphae to yeast in the population profile of *C. albicans* (see Fig 4b). The *S. aureus* population profile at

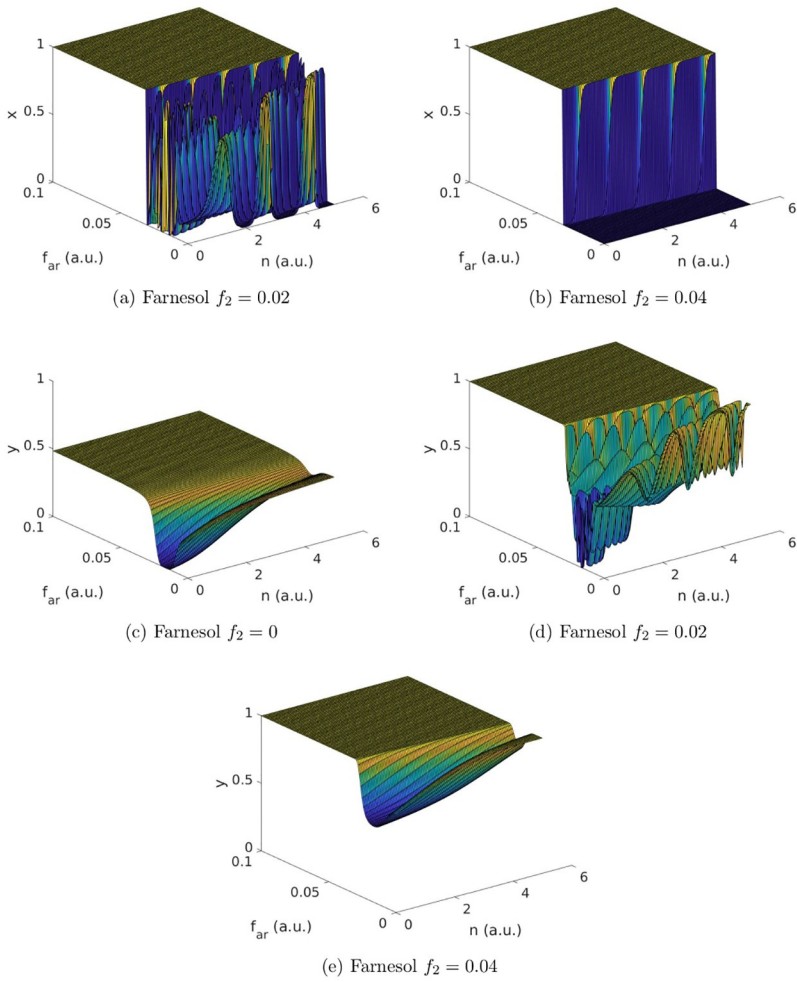

**Fig 4. Simulation results of the case $I_2 \neq 0$.** For selected $f_2$ values the strategy fractions of *C. albicans* ($x$) and *S. aureus* ($y$) are depicted at time step 100 (a.u.) with varying nutrient levels ($n$) and levels of growth change caused by farnesol ($f_{ar}$ and $f_2$). Fig 4a and 4b show selected findings of *C. albicans* while Fig 4c–4e show selected findings of *S. aureus*. The population profiles of *C. albicans* and *S. aureus* differ depending on the number of nutrients ($n$), the levels of change in growth caused by farnesol ($f_{ar}$ and $f_2$) and the gain and loss resulting from the exploitation ($I_2$; see also Fig 5 for the results of the case $I_2 = 0$ for comparison). A summary of the findings is given in Table 4.

moderate levels of $f_2$ ($f_2 = 0.04$) depends on the levels of $f_{ar}$ (see Fig 4e). For high levels of $f_{ar}$, the *S. aureus* population profile consists of cooperators. For low $f_{ar}$ levels, we see a groove towards exploiters reaching $y = 0.56$ at $f_{ar} = 0.016$ for low amounts of nutrients ($n$). This indicates a mixed population profile of cooperators and exploiters. The depth of this groove is nutrient-dependent and levels out for higher $n$ values. For high levels of $f_2$ ($f_2 > 0.06$), the population profile of *S. aureus* consists of cooperators independent of $n$ or $f_{ar}$ (for details see the S1 File).

## The case of $I_2 = 0$

Expectedly, we see the same ESSs for $I_2 = 0$ and $I_2 \neq 0$ at the simulation endpoints. However, in the case of $I_2 = 0$ the ESSs are reached at lower $f_2$ values since the parameter space is shifted

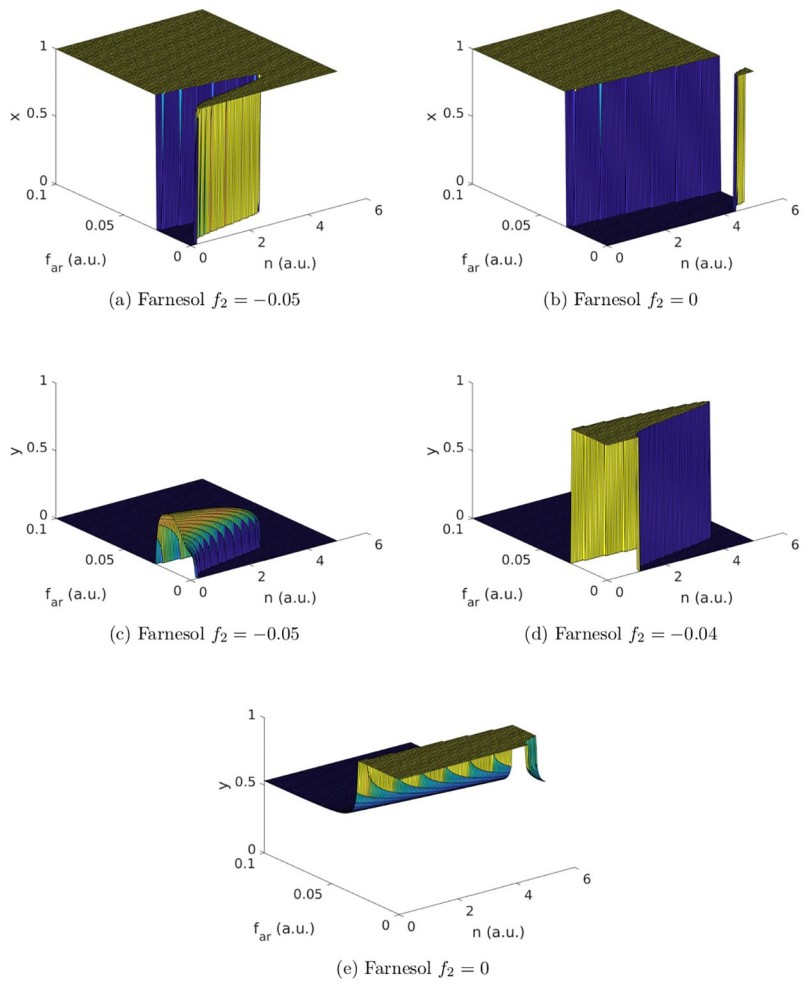

(a) Farnesol $f_2 = -0.05$

(b) Farnesol $f_2 = 0$

(c) Farnesol $f_2 = -0.05$

(d) Farnesol $f_2 = -0.04$

(e) Farnesol $f_2 = 0$

**Fig 5. Simulation results of the case $I_2 = 0$.** For selected $f_2$ values the strategy fractions of *C. albicans* ($x$) and *S. aureus* ($y$) are depicted at time step 100 (a.u.) with varying nutrient levels ($n$) and levels of growth change caused by farnesol ($f_{ar}$ and $f_2$). Fig 5a and 5b show selected findings of *C. albicans* while Fig 5c–5e show selected findings of *S. aureus*. The population profiles of *C. albicans* and *S. aureus* differ depending on the number of nutrients ($n$), the levels of change in growth caused by farnesol ($f_{ar}$ and $f_2$) and the gain and loss resulting from the exploitation ($I_2$; see also Fig 4 for the results of the case $I_2 \neq 0$ for comparison). A summary of the findings is given in Table 4.

due to the omission of $I_2$. Additionally, the system's dynamic behaviour on the path to reaching the EESs differs between the two cases.

When the levels of $f_{ar}$ are high we always observe yeast in the *C. albicans* population independent of $n$ or $f_2$ (see Fig 5a and 5b). For low $f_{ar}$ values, the population profile of *C. albicans* depends on $f_2$ and $n$. For very beneficial (low) $f_2$ values, we find a majority of yeast. However, we also observe a bell-shaped gap of hyphae. Initially a minority, for increasing $f_2$ values this gap of hyphae spreads towards higher $n$ and lower $f_{ar}$ values until hyphae become the dominant form in the *C. albicans* population.

For extremely beneficial (extremely low) values of $f_2$, the *S. aureus* population profile consists of exploiters (for details see the S1 File). With increasing, but still beneficial, $f_2$ values we have to discriminate between high and low values of $f_{ar}$. For high $f_{ar}$ values, the population profile remains unchanged with only exploiters being present. For low $f_{ar}$ values, we see a mixed

**Table 4. The influence of farnesol ($f_2$, $f_{ar}$) and nutritional exploitation ($I_2$) on the outcome of the game.** Summarised are the simulation results for the population profiles of *C. albicans* vs. *S. aureus* of the two case studies $I_2 \neq 0$ in Fig 4 and $I_2 = 0$ in Fig 5 at time step 100 (a.u.) under varying amounts of nutrients ($n$) and levels of growth change caused by farnesol ($f_{ar}$ and $f_2$).

| | $f_2$ | $f_{ar}$ high | $f_{ar}$ low | Figure |
|---|---|---|---|---|
| $I_2 \neq 0$ | beneficial ($f_2 < 0$) | yeast vs. exploiters | yeast vs. exploiters | see the S1 File |
| | unsusceptible ($f_2 = 0$) | yeast vs. mixed population profile | yeast vs. mixed population profile with a nutrient-dependent groove towards exploiters | Fig 4c |
| | low ($f_2 = 0.02$) | yeast vs. cooperators | irregular oscillations in both population profiles | Fig 4a; Fig 4d |
| | moderate ($f_2 = 0.04$) | yeast vs. cooperators | hyphae vs. mixed population profile with a nutrient-dependent groove towards exploiters | Fig 4b; Fig 4e |
| | high ($f_2 > 0.06$) | yeast vs. cooperators | hyphae vs. cooperators | see the S1 File |
| $I_2 = 0$ | very beneficial ($f_2 < -0.05$) | yeast vs. exploiters | yeast with a bell-shaped gap of hyphae vs. exploiters and a mixed population profile with a raising towards cooperators | Fig 5a; Fig 5c |
| | beneficial ($f_2 = -0.04$) | yeast vs. exploiters | yeast with a bell-shaped gap of hyphae vs. exploiters with a bell-shaped raising of cooperators | similar to Fig 5a (see the S1 File); Fig 5d |
| | unsusceptible ($f_2 = 0$) | yeast vs. mixed population profile | hyphae except for a minority of yeast for very high $n$ values vs. cooperators with a slope towards a mixed population profile for very high $n$ values | Fig 5b; Fig 5e |
| | low, moderate & high ($f_2 > 0$) | yeast vs. cooperators | hyphae vs. cooperators | similar to Fig 4b see the S1 File |

population profile with a bell-shaped raising towards cooperators (see Fig 5c). This raising towards cooperators for low $f_{ar}$ values increases with increasing $f_2$ values. Within a few simulation steps, the raising of a mixed population profile turns into a bell-shaped plateau of cooperators (see Fig 5d).

This dynamic changes for unsusceptible *S. aureus* strains ($f_2 = 0$). For high $f_{ar}$ values, we see a mixed population profile of cooperators and exploiters. For low $f_{ar}$ values, we see a majority of cooperators except for very high amounts of nutrients ($n$) where again a mixed population profile occurs (see Fig 5e).

For low, moderate and highly susceptible *S. aureus* strains ($f_2 > 0$) we only find cooperators independent of $n$ or $f_{ar}$ (for details see the S1 File).

## Discussion

The human microbiome is a topic of great current interest [67–70]. Our paper contributes to its analysis by studying the interactions of the fungus *C. albicans* and the bacterium *S. aureus*, both of which can reside in the human body. Mixed-species biofilms of *C. albicans* and *S. aureus* are usually thought to be beneficial to both organisms [5, 11, 12]. However, some studies suggest that under certain conditions like nutrient sparsity, the interactions of *C. albicans* and *S. aureus* can become antagonistic [11, 15]. A possible role in orchestrating the interactions between *C. albicans* and *S. aureus* may be played by *C. albicans*' quorum-sensing molecule farnesol [7, 22]. For this reason, farnesol is proposed as a medical adjuvant to shift the balance of the mixed-species biofilm to more favourable conditions for the patient [57].

In this paper, we answer the questions if mixed-species biofilms of *C. albicans* and *S. aureus* are always mutually beneficial; whether the suggested scavenging of *C. albicans* hyphae by *S. aureus* has an impact on the biofilm dynamic; how the state of the biofilm can be changed and how artificially added farnesol is impacting the state of the biofilm.

For this, we used mathematical modelling to study the interactions of *C. albicans* and *S. aureus* in a mixed-species biofilm. Applying EGT we studied the Nash equilibria and ESSs of the game between these two pathogens. Using replicator equations we investigated the population dynamics of the *C. albicans*-*S. aureus* biofilm. To analyse different states of the biofilm we

allowed for different parameter levels and modelled the differences between nutrient abundance and sparsity of the surroundings. To test the influence of artificially added farnesol and the suggested feeding of *C. albicans'* hyphae by *S. aureus*, we modelled and compared two cases with and without nutritional exploitation by *S. aureus*. Although our model is not directly based on quantitative experimental data, it elucidates the system behaviour under various conditions and provides testable hypotheses.

For future studies, one could also use Lotka-Volterra equations as an alternative approach to replicator equations [71]. Extending our study to different pHs in different human body niches or medical device environments is a promising avenue for future research. Investigating the nuanced impact of pH variations on the dynamics of mixed-species biofilms involving *C. albicans* and *S. aureus* could offer invaluable insights into the adaptability and behaviour of these biofilms in specific contexts. Furthermore, refining our model by analysing the role of the extracellular polymeric substance (EPS) and dissecting the overarching parameters, such as $r$ and $n$ into distinct components $r_1, r_2, \ldots$ and $n_1, n_2, \ldots$ might enable a detailed exploration of the role played by individual factors. This granular analysis could shed light on the specific influence of, for example, antibiotic and antifungal drug presence or the contributions of the biofilm matrix to resistance mechanisms and biofilm dynamics. However, given the complexity of the current model and the risk of over-fitting, a balanced approach is crucial.

### Are mixed-species biofilms of *C. albicans* and *S. aureus* mutually beneficial?

Contrary to common assumptions, we confirm the hypothesis that under certain conditions, mixed-species biofilms are not universally beneficial. Instead, different Nash equilibria occur depending on encountered conditions (for example, varying farnesol levels, either produced by *C. albicans* or artificially added), including antagonism.

Studying the replicator equations of the game we showed that mixed-species biofilms of *C. albicans* and *S. aureus* are not necessarily mutually beneficial. Instead, we found that five different pairings of population profiles (the fixed points of the system), each representing a unique state of the biofilm, can arise depicting the whole spectra from cooperation to antagonism. Four of these strategy pairings (yeast—cooperation, yeast—exploitation, hyphae—cooperation and hyphae—exploitation) are Nash equilibria in pure strategies being evolutionary stable (see Table 3). Under certain conditions, a fifth Nash equilibrium in mixed strategies with oscillations in strategy fractions within the populations can occur. However, these oscillations are not evolutionary stable, with small variations in the biofilm conditions changing the population profiles of both organisms.

### Does the suggested scavenging of *C. albicans* hyphae by *S. aureus* have an impact on the biofilm?

When analysing the suggested feeding of *C. albicans'* hyphae by *S. aureus*, we found that the overall outcome of the game (that is the possible pairings of ESS) does not change when considering feeding ($I_2 \neq 0$) or non-feeding ($I_2 = 0$). In both cases, we see the same qualitative behaviour of the dynamics, only reached with different parameter sets, as the course of the curve and the time needed to reach a fixed point are parameter-dependent (see Fig 2). That means that the scavenging of *C. albicans'* hyphae by *S. aureus* does not influence if the state of the biofilm becomes antagonistic but rather at what point in time. Even without *S. aureus* feeding of *C. albicans'* hyphae the state of the biofilm can become antagonistic. It is worth noting that because of the parameter-dependencies, the strategy fractions of *C. albicans* and *S. aureus* before reaching an ESS differ between the two cases but also between different parameter sets

within each case (see Figs 4 and 5). The state of the biofilm is especially dependent on the available nutrients provided by the host ($n$ and $I_1$) and the levels of change in growth caused by farnesol ($f_2$ and $f_{ar}$). Interestingly and somehow surprisingly, *C. albicans'* farnesol production cost ($f_1$) does not seem to have an impact on the outcome of the game.

### How can the state of the biofilm be changed?

The best chances for medical treatment and for the immune response to get the biofilm under control are when the biofilm state shows the strategy pairings of either yeast—cooperation or yeast—exploitation. In both game scenarios, both pathogens grow at their basic growth rates, not profiting from heightened resistance. This will facilitate medical treatment and increase the success of the host immune response against the biofilm. As a consequence, a solution to actively shift the state of the biofilm at any given time to strategy pairings of yeast—exploiters or yeast—cooperators even under conditions favouring hyphae, is needed.

Our studies indicate that one such solution could be to lower the gain in nutrients by invading the host ($I_1$) to a critical threshold. *C. albicans* is likely to choose its yeast strategy facing low levels of nutrients provided by the host. However, this approach is only feasible in *in vitro* experiments. In *in vivo* treatments, this approach is not applicable. Another solution approach could be to dampen the increase in resistance ($r$). This would favour *C. albicans'* yeast strategy but also requires more research.

More promising seems the approach to alter the state of the biofilm using the effect of farnesol on the system. In theory, there are multiple options for influencing the system through farnesol. As farnesol influences *S. aureus'* growth rate, one could think of altering the bacterium's sensitivity to the quorum sensing molecule through medication. Depending on the farnesol level of the system *S. aureus'* growth rate is either enhanced or decreased concerning the tolerance against antibacterials. Low levels of farnesol are increasing the bacterium's growth. However, passing a certain threshold, farnesol decreases the bacterium's growth [38]. By lowering *S. aureus'* threshold to farnesol, one could limit the growth of the biofilm. However, this approach risks a shift in the dynamics of the biofilm towards the cooperation of the two pathogens. *S. aureus* chooses cooperation to avoid high levels of farnesol, while *C. albicans* chooses its hyphae strategy as the level of farnesol is low. This shift would lead to a severe course of disease for the patient.

To avoid the cooperation of the two pathogens, one could heighten the farnesol level in the system. This approach would use the fact of decreased growth rates in *S. aureus*, passing a threshold in farnesol level, while at the same time taking into account the impact of farnesol on *C. albicans'* morphological switch. This would result in *S. aureus* choosing its cooperation strategy and *C. albicans* choosing its yeast strategy. To further investigate this approach we studied the effect of artificially added farnesol on the system under varying nutrient concentrations and levels of change in growth caused by farnesol.

### How is artificially added farnesol impacting the state of the biofilm?

Our research indicates that artificially added farnesol has a major impact on the dynamics of the game (see Table 4). The state of the biofilm is hereby highly dependent on the impact of released farnesol on *S. aureus* ($f_2$) and the amount of artificially added farnesol ($f_{ar}$). If one decides to use farnesol as a treatment the level of artificially added farnesol needs to be sufficiently high. Otherwise, cooperation between the two pathogens with its negative implications on the host can occur. However, as a quorum sensing molecule, farnesol might not only influence the mixed-species biofilm but also have an effect on other microbiota in the environment. Further studies are needed to assess the side effects and risks of high doses of farnesol on the

patient. Simply decreasing the amount of added farnesol on the other side does not have the desired effect. Our studies show that too low levels of added farnesol allow for undesired strategy pairings, enabling cooperation, resulting again in negative implications on the host.

## Conclusion

Our study contributes to the understanding of the complex dynamics within mixed-species biofilms, in particular, shedding light on the interactions between *C. albicans* and *S. aureus* and their responses to various environmental factors. The study identifies different states of the mixed-species biofilm, offering opportunities to explore interventions that shift the equilibrium towards more favourable conditions for medical treatments and host immune responses.

In our study, we identify non-universal benefits. Against common belief [5–7, 11, 12] we showed that mixed-species biofilms of *C. albicans* and *S. aureus* are not necessarily mutually beneficial. Instead, we found that the whole spectra from cooperation to antagonism is depicted in the choice of strategy pairings. Recognizing the nuanced and context-dependent nature of these interactions is crucial for devising targeted antibiofilm strategies. We showed that the suggested scavenging of *C. albicans*' hyphae by *S. aureus* [15] does not influence the overall outcome of the game. Even without *S. aureus* feeding of *C. albicans*' hyphae the state of the biofilm can become antagonistic, with both players no longer cooperating with each other.

Our study helps in understanding farnesol dynamics. Our focus on the quorum-sensing molecule farnesol provides insights into its impact on biofilm dynamics. The state of the biofilm is especially dependent on the available nutrients provided by invading the host ($I_1$) and the levels of change in growth caused by farnesol ($f_2$ and $f_{ar}$). It is noteworthy that *C. albicans*' farnesol production cost ($f_1$) seems not to influence the outcome of the game. We further confirmed that artificially added farnesol has a major impact on the dynamics of the game. However, if one decides to use farnesol as a treatment the level of artificially added farnesol needs to be sufficiently high. Otherwise, cooperation between the two pathogens with its negative implications on the host can occur. Understanding how artificially added farnesol influences the game between *C. albicans* and *S. aureus* opens avenues for exploring its potential therapeutic use in manipulating biofilm states. Fine-tuning the dosage and administration could enhance its efficiency in controlling biofilm states. These insights may guide the development of tailored antibiofilm strategies based on the specific conditions encountered, allowing for more effective and personalized treatment approaches. Manipulating the pathways could offer strategies to disrupt biofilm formation and potentially augment existing antimicrobial therapies. By depicting how factors such as available nutrients, microbial interactions, and specific molecules impact biofilm dynamics, our study offers a framework for identifying critical targets for intervention.

## Supporting information

**S1 File. Additional simulation results.**
(PDF)

**S2 File. Stability analysis of the five fixed points of the game.**
(PDF)

## Author Contributions

**Conceptualization:** Sybille Dühring.

**Data curation:** Sybille Dühring.

**Formal analysis:** Sybille Dühring.

**Funding acquisition:** Stefan Schuster.

**Investigation:** Sybille Dühring.

**Methodology:** Sybille Dühring.

**Project administration:** Sybille Dühring.

**Resources:** Sybille Dühring, Stefan Schuster.

**Software:** Sybille Dühring.

**Supervision:** Stefan Schuster.

**Validation:** Sybille Dühring.

**Visualization:** Sybille Dühring.

**Writing – original draft:** Sybille Dühring.

**Writing – review & editing:** Sybille Dühring, Stefan Schuster.

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
