## [Decision Letter · Decision Letter 0]

30 Oct 2023

PONE-D-23-29022Studying mixed-species biofilms of *Candida albicans* and *Staphylococcus aureus* using Evolutionary Game TheoryPLOS ONE

Dear Dr. Dühring,

Thank you for submitting your manuscript to PLOS ONE. After careful consideration, we feel that it has merit but does not fully meet PLOS ONE’s publication criteria as it currently stands. Therefore, we invite you to submit a revised version of the manuscript that addresses the points raised during the review process.

We look forward to receiving your revised manuscript.

Kind regards,

Geelsu Hwang, Ph.D.

Academic Editor

PLOS ONE

Journal Requirements:

We acknowledge support by the Deutsche Forschungsgemeinschaft (DFG) within the CRC/Transregio 124 ‘Pathogenic fungi and their human host’ (project number 210879364), subproject B1. The funders had no role in study design, data collection and analysis, decision to publish, or preparation of the manuscript.

Reviewers' comments:

Reviewer's Responses to Questions

**Comments to the Author**

1. Is the manuscript technically sound, and do the data support the conclusions?

Reviewer #1: Yes

Reviewer #2: Yes

2. Has the statistical analysis been performed appropriately and rigorously? 

Reviewer #1: Yes

Reviewer #2: Yes

3. Have the authors made all data underlying the findings in their manuscript fully available?

Reviewer #1: Yes

Reviewer #2: Yes

4. Is the manuscript presented in an intelligible fashion and written in standard English?

Reviewer #1: Yes

Reviewer #2: Yes

5. Review Comments to the Author

Reviewer #1: Author prosed a nice study. I have following comment;

1: Besides considering the available nutrients and the quorum sensing molecule farnesol for making the model. Author should also consider the pH of the environment as an important factor. Mostly these two microbe present in symbiotic in the oral cavity. In the oral cavity there is drop of the pH to acidic. Then how this drop of pH influence the mixed biofilm with respect to the morphology, physiology and antimicrobial resistance.

2: How the mixed biofilm of S. aureus and C. albicans can be fitted in the proposed model in the presence of the antibiotic and antifungal drugs.

3: What about the EPS of biofilm matrix contribution by these microbes. On what ratio?

4: How this study will benefit to develop antibiofilm strategy.

Reviewer #2: A very unusual approach, definitely - the article deserves wide publicity. An excellent statistical work that has found its application in microbiology. Description of different strategies for the development of mixed biofilms. I would like to ask the author: what practical application of this work does he see?

6. PLOS authors have the option to publish the peer review history of their article (what does this mean?). If published, this will include your full peer review and any attached files.

Reviewer #1: No

Reviewer #2: **Yes: **Nadezhda Sachivkina

---

## [Author Response · Author response to Decision Letter 0]

14 Dec 2023

Subject: Response to Reviewers' and Editor's Comments on Manuscript Submission PONE-D-23-29022 - EMID:6e14a60959737897/Studying mixed-species biofilms of Candida albicans and Staphylococcus aureus using Evolutionary Game Theory

Dear Dr. Hwang,

We sincerely appreciate the time and effort invested by you and the esteemed reviewers in evaluating our manuscript titled "Studying mixed-species biofilms of Candida albicans and Staphylococcus aureus using Evolutionary Game Theory" for potential publication in PLOS ONE. Your insights and recommendations have been invaluable in refining our work. We have carefully considered each suggestion and comment provided by the reviewers and you and have incorporated them into the revised manuscript. We have attached a revised version of the manuscript. Please find our point-by-point responses to the reviewers' and your comments below.

 Journal Requirements:

 1. "[...] Please ensure that your manuscript meets PLOS ONE's style requirements, including those for file naming. [...]": 

 Thank you very much for providing the formatting guidelines. We have used the Latex template provided by PLOS ONE to format the manuscript. We have also checked against the guidelines provided and hope to adhere to the journal standards. If there are any specific areas that need to be adapted, or if there are particular instructions you'd like us to follow, we would greatly appreciate your guidance. Your specific details will help us to promptly address any necessary changes to ensure full compliance with PLOS ONE's style requirements.

 2. "[...] Please provide an amended statement that declares all the funding or sources of support (whether external or internal to your organization) received during this study [...]": 

 Thank you for updating our amended Funding Statement in the online submission form on our behalf. Our amended statement is: "We acknowledge support by the Deutsche Forschungsgemeinschaft (DFG) within the CRC/Transregio 124 ‘Pathogenic fungi and their human host’ (project number 210879364). The funders had no role in study design, data collection and analysis, decision to publish, or preparation of the manuscript. There was no additional (internal and external) funding received for this study."

 3. "Upon re-submitting your revised manuscript, please upload your study’s minimal underlying data set as either Supporting Information files or [...]: 

 Thank you for updating our Data Availability statement on our behalf. All the data is now in the manuscript. The model and parameter set to reproduce our simulation results are given in the Methods chapter (page 4-7) and the Results chapter (Table 2 on page 9 and page 11 paragraph 1 line 319-325) of the paper. To include all data we added another Supporting Information file named S2.pdf with the figures of the case study previously omitted from the paper (case I_2 unequal 0 for beneficial and high f_2 values as well as case I_2=0 for beneficial and low/moderate/high f_2 values). The results are either trivial or very similar to the figures already shown in the manuscript, depicting the same qualitative behavior. We therefor think there are most suited as Supporting Information. We updated paragraph 3 on page 8, Table 4 on page 12, paragraph 1, 4 and 7 on page 13 as well as paragraph 2 on page 14 in the manuscript to reference the new S2.pdf file. We further added the S2 File in the list of supporting information on page 21. We have found two minor typing errors (100 instead of 1000 simulation steps as well as two values for f2 in Table 4) and have corrected them in the whole manuscript. Figures 4 and 5 have been adjusted accordingly. These adjustments have no effect on the presentation of the figures or their discussion. The qualitative behaviour and the results remain consistent and are not affected by these restatements.

 Review Comments:

 Reviewer #1:

 1: "Author should also consider the pH of the environment as an important factor. Mostly these two microbe present in symbiotic in the oral cavity. In the oral cavity there is drop of the pH to acidic. Then how this drop of pH influence the mixed biofilm with respect to the morphology, physiology and antimicrobial resistance.": 

 We agree with the reviewer that the pH of the environment is an important factor. Certainly, addressing the influence of pH on the mixed biofilm's morphology, physiology, and antimicrobial resistance is an intriguing consideration. While integrating pH as a factor could potentially enhance the depth of our study, the feasibility and potential complexity associated with its inclusion must be carefully considered. Given the complexity of the current model and the risk of over-fitting, a balanced approach is crucial. Extending the study to incorporate pH requires significant additional effort and might introduce complexities that could overshadow the primary focus of our research. Furthermore, the interaction between the pathogens extends beyond the oral cavity (see Kong et al. 2016, Kean et al. 2017, Lohse et al. 2018). We added the potential implications of pH variations on the interactions between these pathogens in diverse environments in the Discussion section. For this we have added paragraph 4 on page 14. This approach would acknowledge the significance of pH variations without compromising the current depth and focus of our study. This approach also avoids the risk of potential over-fitting and increased complexity, which does not necessarily lead to significantly better results. However, recognizing the importance of pH variation as a potential avenue for future investigation allows us to maintain a broad perspective without compromising the integrity of our current research. 

 2: "How the mixed biofilm of S. aureus and C. albicans can be fitted in the proposed model in the presence of the antibiotic and antifungal drugs.": 

 We agree with the reviewer that the presence of antibiotic and antifungal drugs is an important factor. In our study, we have indirectly addressed the interaction within the mixed biofilm of S. aureus and C. albicans in the presence of such drugs by incorporating a general factor, 'r', to signify the increased resistance observed when these pathogens coexist. This factor encapsulates the combined impact on resistance, encompassing both the influence on drug resistance and the heightened resistance to the immune system. 

 One could consider a potential extension of the model by delineating resistance increments (r1, r2, ...) to differentiate the resistance increments between the immune system and drugs. However such an extension could artificially inflate the complexity of our current model. Given the complexity and depth of the proposed model extension, incorporating distinct resistance increments for the immune system and drug resistance might introduce unnecessary intricacies without substantially enhancing the core findings of our research. Maintaining a balance between comprehensiveness and model simplicity is vital to ensure clarity and relevance in our study. Additionally, incorporating new parameters, such as r1, r2, ..., would necessitate comprehensive data to run accurate simulations. This level of granularity exceeds the intended scope of our study and might potentially obscure the primary focus.

 For the purposes of our current study, we believe retaining the generalized factor 'r' appropriately captures the increased resistance observed within the mixed biofilm context. However, exploring the population dynamics of both pathogens under the influence of specific drugs presents an intriguing and promising outlook for future research. Recognizing its significance, we have included paragraph 4 on page 14. 

 3: "What about the EPS of biofilm matrix contribution by these microbes. On what ratio?": 

 We thank the reviewer for inquiring about the role of the extracellular polymeric substance (EPS) within the biofilm matrix of S. aureus and C. albicans and its impact within our proposed model. The biofilm matrix with its EPS indeed plays a critical role in biofilm formation, resistance to antimicrobial agents, and contributes to the unique attributes and virulence of biofilms. 

 In our study, we've indirectly examined the biofilm matrix's impact on the interactions within mixed biofilms of S. aureus and C. albicans. While acknowledging the crucial role of biological details, including EPS contribution, we've balanced the depth of biological detail with the focus on our overarching model to ensure clarity. The factor 'r' i.e. addresses increased resistance in the mixed biofilm, encapsulating various elements that contribute to this augmented resistance, potentially including effects from the biofilm matrix like EPS. In addition the parameter 'n' i.e. summarizes all available nutrients including potential nutrient reservoirs of the biofilm matrix. This approach aligns with our goal of understanding interactions within mixed biofilms while managing complexity. 

 In future studies one could further specify each factor that plays a vital role in the system by further splitting parameters i.e. r into r1, r2, ... and n into n1,n2,... However the approach brings its complications i.e. over-fitting as addressed in the response to point 1 and 2 of reviewer 1. Expanding into specific ratios of EPS contribution by S. aureus and C. albicans introduces complexities that could overshadow our model's primary objectives. We have now added paragraph 4 on page 14 to point out future study approaches and mention the EPS therein. 

 Reviewer #1 4. and Reviewer #2:

 "How this study will benefit to develop antibiofilm strategy." and "[...] What practical application of this work does he see?": 

 We appreciate the reviewer’s inquiry regarding the practical implications and potential applications of our research in developing antibiofilm strategies. 

 In our study we identify non-universal benefits. Contrary to common assumptions, our research reveals that mixed-species biofilms are not universally beneficial for all participants under all conditions. Recognizing the nuanced and context-dependent nature of these interactions is crucial for devising targeted antibiofilm strategies. 

 We explore different treatment opportunities. The study identifies different states of the mixed-species biofilm, offering opportunities to explore interventions that shift the equilibrium towards more favorable conditions for medical treatments and host immune responses. These insights may guide the development of tailored antibiofilm strategies based on the specific conditions encountered, allowing for more effective and personalized treatment approaches. 

 To strengthen the argument of our paper we have included paragraph 1 and 4 as well as 2 sentences to paragraph 2 and 5 sentences to paragraph 3 of the conclusion section of our paper on page 16 and 17.

Once again, we extend our gratitude for your time and constructive feedback. We look forward to hearing from you regarding the revised manuscript.

Sincerely,

Sybille Dühring

sybille.duehring@uni-jena.de

References:

Eric F Kong et al. “Commensal Protection of Staphylococcus aureus against

Antimicrobials by Candida albicans Biofilm Matrix”. In: mBio 7.5 (2016),

e01365–16.

Ryan Kean et al. “Candida albicans mycofilms support Staphylococcus aureus

Colonization and Enhances Miconazole Resistance in Dual-Species Interactions”.

In: Frontiers in Microbiology 8 (2017), p. 258.

Matthew B Lohse et al. “Development and regulation of single-and multi-species

Candida albicans biofilms”. In: Nature Reviews Microbiology 16.1 (2018),

pp. 19–31. doi: 10.1038/nrmicro.2017.107.

---

## [Decision Letter · Decision Letter 1]

3 Jan 2024

Studying mixed-species biofilms of *Candida albicans* and *Staphylococcus aureus* using Evolutionary Game Theory

PONE-D-23-29022R1

Dear Dr. Dühring,

We’re pleased to inform you that your manuscript has been judged scientifically suitable for publication and will be formally accepted for publication once it meets all outstanding technical requirements.

Kind regards,

Geelsu Hwang, Ph.D.

Academic Editor

PLOS ONE

\\Reviewers' comments:

Reviewer's Responses to Questions

**Comments to the Author**

1. If the authors have adequately addressed your comments raised in a previous round of review and you feel that this manuscript is now acceptable for publication, you may indicate that here to bypass the “Comments to the Author” section, enter your conflict of interest statement in the “Confidential to Editor” section, and submit your "Accept" recommendation.

Reviewer #1: All comments have been addressed

2. Is the manuscript technically sound, and do the data support the conclusions?

Reviewer #1: Yes

3. Has the statistical analysis been performed appropriately and rigorously? 

Reviewer #1: Yes

4. Have the authors made all data underlying the findings in their manuscript fully available?

Reviewer #1: Yes

5. Is the manuscript presented in an intelligible fashion and written in standard English?

Reviewer #1: Yes

6. Review Comments to the Author

Reviewer #1: Congratulations to all the authors for their great work. The author has addressed all the raised comments point by point.

7. PLOS authors have the option to publish the peer review history of their article (what does this mean?). If published, this will include your full peer review and any attached files.

Reviewer #1: No

---

## [Editor Report · Acceptance letter]

7 Feb 2024

PONE-D-23-29022R1 

PLOS ONE

Dear Dr. Dühring, 

I'm pleased to inform you that your manuscript has been deemed suitable for publication in PLOS ONE. Congratulations! Your manuscript is now being handed over to our production team.

Kind regards, 

on behalf of

Dr. Geelsu Hwang 

Academic Editor

PLOS ONE